# Accuracy Boosters: Epoch-Driven Mixed-Mantissa Block Floating-Point for DNN Training

## Abstract

The unprecedented growth in DNN model complexity, size and the amount of training data have led to a commensurate increase in demand for computing and a search for minimal encoding. Recent research advocates Hybrid Block Floating-Point (HBFP) as a technique that minimizes silicon provisioning in accelerators by converting the majority of arithmetic operations in training to 8-bit fixed-point. In this paper, we perform a full-scale exploration of the HBFP design space including minimal mantissa encoding, varying block sizes, and mixed mantissa bit-width across layers and epochs. We propose *Accuracy Boosters*, an epoch-driven mixed-mantissa HBFP that uses 6-bit mantissa only in the last epoch and converts $99.7\%$ of all arithmetic operations in training to 4-bit mantissas. Accuracy Boosters enable reducing silicon provisioning for an HBFP training accelerator by $16.98\times$ as compared to FP32, while preserving or outperforming FP32 accuracy.

## 1 Introduction

Improvements in Deep Neural Network (DNN) algorithms over the past decade have led to an unprecedented growth in model complexity and dataset size and consequently the required computational resources to train DNN models. One of the largest DNN models (GPT-3) (Brown et al., 2020) has 175 billion parameters and requires $3.14 \times 10^{23}$ FLOPs to train. With the slowdown in Moore's law, researchers and vendors have begun to search for ways to improve the arithmetic density of the underlying hardware platforms. Narrow bit-width (with lower precision) number formats (Wang & Kanwar, 2019; Micikevicius et al., 2018; Sun et al., 2019; Mellempudi et al., 2019; Sun et al., 2020) have emerged as a promising approach to increase arithmetic density, as well as, reduce the required operand storage and communication bandwidth while maintaining high accuracy for training.

Many have tried fixed-point formats, often used in inference, to further reduce silicon logic complexity for arithmetic (Courbariaux et al., 2015a; Hubara et al., 2016; Rastegari et al., 2016; Dettmers et al., 2022a;b). Fixed-point formats, unfortunately, dramatically suffer from a limited range in numerical representation especially for arithmetic in backward propagation. As such, researchers have tried mixed-precision training to trade off accuracy for efficiency (Zhang et al., 2021; Fu et al., 2020; 2021).

Recently there have been several proposals for block floating point (Köster et al., 2017; Das et al., 2018; Zhang et al., 2021), a numerical encoding that groups a block of mantissas which rely on only fixed-point arithmetic with a single exponent. Block floating point asympototically approaches the arithmetic density of fixed point with larger block sizes and naturally lends itself well to mixed-precision hardware where a block with the same number of exponent bits can have a fixed-point datapath which is bitsliced for various multiples of mantissa bit encodings (e.g., the same today's CPU cores implement SIMD). While block floating point has been promising in use for inference (e.g., Microsoft Floating Point (Darvish Rouhani et al., 2020)), most proposals to train with block floating point have either failed to reach its full potential by requiring small blocks and/or just fall short of reaching FP32 accuracy.

One specific proposal, Hybrid Block Floating Point (HBFP) (Drumond et al., 2018), uses a mixed-precision format where the dominant fraction of training which is the dot products happens in block floating point (e.g., convolutions, matrix multiplications, outer products), and higher precision (e.g.,

FP32) is used for other less frequent operations requiring larger numerical ranges (e.g., activations, regularizations). HBFP simultaneously offers the high accuracy of floating point and the superior hardware density of fixed point, delivering up to $8.5\times$ higher throughput than FP16 with $2\times$ more compact models (Drumond, 2020). Prior work on HBFP only presented a preliminary analysis of the design space for power-of-two mantissa bit widths (e.g., 2, 4, 8-bit mantissas).

In this paper, we make the observation that the parameter space for HBFP is quite rich presenting several opportunities for further improving efficiency and density in hardware platforms. First, custom accelerators can support non-power-of-two numerical formats; and minimizing the number of bits improves operand storage and communication linearly, and arithmetic logic quadratically. Second, there is an interplay between the block size and the number of mantissa bits, allowing for an overall denser numerical format with smaller blocks while maintaining high accuracy. Finally, HBFP allows for mixed-mantissa block floating point encodings. Prior work studies training with various HBFP formats in isolation; however, the design space of mixed-mantissa HBFP is yet to be explored.

To minimize the number of bits in HBFP, we explore the interplay between the block size and the number of mantissa bits. We show that HBFP with six or more mantissa bits has no sensitivity to block size. While HBFP with a smaller number of mantissa bits is sensitive to block size, these configurations do not result in enough accuracy even with the smallest blocks and require additional methods to increase accuracy. Accuracy Boosters further minimizes the number of bits and enables training with 4-bit mantissas. Our method improves epoch-wise mixed-precision training by introducing high precision to the training process only at the last epoch. The main contributions of this paper are as follows:

- We show that HBFP6 is the smallest HBFP format achieving competitive accuracies.

- We enable HBFP5 training for smaller models (e.g., ResNet20) by using small block sizes and for larger models (e.g., DenseNet40) by keeping the first and last layers in FP32.

- We improve the silicon area- and energy-efficiency of training without significant accuracy loss by performing a large fraction of epochs in low precision for CNN and Transformer models. We show that for a few models, our method even outperforms FP32 training.

## 2   Why minimize HBFP?

HBFP is a mixed-precision training technique that brings area- and energy-efficient fixed-point arithmetic into DNN training. DNN training traditionally uses floating-point operations because training algorithms require high precision. However, floating-point hardware occupies a larger silicon area and consumes more energy than the fixed-point arithmetic due to exponent management and normalization. HBFP alleviates the need for floating-point operations by employing fixed-point arithmetic for dot products, the most common type of operations in DNN training. Furthermore, it assigns exponents to fixed-point tensors to emulate the dynamic range provided by floating-point to maintain the training accuracy. In other words, HBFP achieves fixed-point efficiency in DNN training with floating-point accuracy.

We argue that employing both smaller mantissa bit widths and larger block sizes are the keys to improving HBFP hardware efficiency. Prior results show that, to the first order, minimizing the number of bits in fixed-point arithmetic reduces operand storage and memory bandwidth linearly and multiplication power and area quadratically (Gholami et al., 2021). As a result, HBFP hardware efficiency increases with the reduced number of mantissa bits due to the large fraction of fixed-point operations. We also note that the hardware area and energy expenditure of HBFP accelerators is determined by the number of mantissa bits and the block size because the overhead of the exponent bits is negligible. Therefore, we work with 10-bit exponents as in prior work (Drumond et al., 2018), and explore the HBFP design space by varying the mantissa bit width and the block size. Our experiments with different mantissa bit widths give rise to a re-configurable DNN accelerator for mixed-mantissa HBFP. Furthermore, we demonstrate that using smaller block sizes reduces the number of fixed-point operations in dot products causing an increase in the area overhead. In line with these observations, we establish decreasing the mantissa bit widths and increasing the block sizes as design guidelines for HBFP-based accelerators.

To validate our guidelines, we compare the area improvements of HBFP formats of different mantissa bit widths and block sizes, by synthesizing the arithmetic units using the Synopsys Design Compiler and TSMC 28nm technology. We determine the costs of individual design components and combine them using the mathematical equations that govern the entire accelerator structure. For FP32 dot product units of size $N$, we estimate the hardware cost as the sum of the cost of $N-1$ FP32 adders, $N$ FP32 multipliers, one FP32 accumulator (adder), and one floating-point activation unit. For HBFP dot product units, we estimate the hardware cost as the sum of the cost of $N-1$ fixed-point adders, $N$ fixed-point multipliers, one FP32 accumulator (adder), one floating-point activation unit, and one adder for signed exponents. We also add the costs of conversions between FP32 and fixed-point numbers by synthesizing the converter blocks. In Figure 1, the nomenclature HBFP$m$ refers to an HBFP configuration with mantissa bit width $m$. We observe that a reduction in the mantissa bits alone (when the block size is fixed to 256) results in a $10\times$ and $17\times$ reduction in area and power using HBFP6 and HBFP4 respectively relative to FP32.

Being a numerical format originally proposed for DNN training, HBFP is also capable of inference, a task that is inherently easier than training. We observe that HBFP8 and HBFP6 with a block size of 576 are suitable for inference for the most popular CNN models (i.e., ResNet, MobileNet, VGG, ShuffleNet) without any accuracy loss. Furthermore, MSFP, an HBFP-like numerical format that uses signed magnitude arithmetic, was shown to achieve superior inference performance to out-of-the-box INT8 quantization (Darvish Rouhani et al., 2020). These findings suggest that HBFP is a versatile numerical format for both training and inference, paving the way to a unified accelerator capable of efficient, accurate training and high-performance inference.

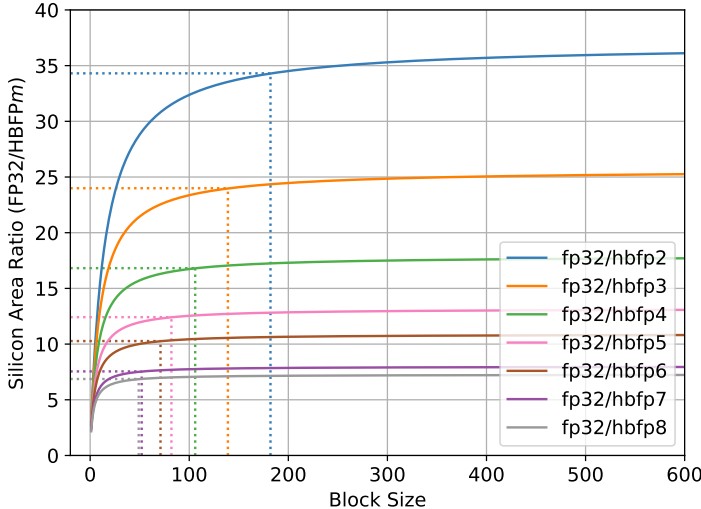

Figure 1: The silicon area ratio between FP32 and HBFP with various block sizes. The dotted lines show the points when the design achieves 95% of the maximum hardware benefit.

The rest of this paper provides a step-by-step explanation of how to minimize the number of bits in HBFP with the help of mixed-precision training while maintaining or outperforming FP32 accuracy.

## 3  MINIMIZING HBFP

In this section, we describe the steps to follow to minimize HBFP, that is to bring smaller mantissa bit widths and larger block sizes into training. We investigate the interplay between the number of mantissa bits and the block size, and also use an existing technique to recover the data loss, keeping the first and last layers of CNNs in FP32. We leverage and elaborate on the insight that not all epochs have the same effect on the DNN model accuracy, and introduce the Accuracy Boosters method.

For the rest of this paper, we use the notation HBFP$(m, N)$, for an HBFP format with a block size of $N$, mantissa bit-width $m$ and 10-bit exponent. For a tensor encoded in this format, we have a

mantissa tensor consisting of $N$-many elements and a shared exponent, both of which using two's complement arithmetic.

## 3.1 THE SMALLEST HBFP CONFIGURATION

Prior work has shown that $\text{HBFP}(8, 576)$ is the smallest HBFP format with power-of-two mantissa bits that can train a wide range of models like CNNs, LSTMs, and BERT with FP32 accuracy (Drumond, 2020). They also show that HBFP8 shows almost no sensitivity to the block size, meaning that arbitrarily large block sizes yield similar accuracy, within $0.5\%$ of the FP32 baseline. However, they did not explore the entire parameter space and focused only on power-of-two configurations (HBFP8 and HBFP4). While power-of-two-bit numbers align naturally with the memory structure and encode matrices in a tightly-packed way, non-power-of-two-bit mantissas can improve the arithmetic density even further, as studied by Darvish Rouhani et al. (2020).

Considering that a non-power-of-two format is feasible, we claim that HBFP6 achieves FP32 accuracy providing around $1.5\times$ silicon area gain over HBFP8. Among 16-bit encodings, FP16 without mixed-precision training does not reach FP32 accuracy (Micikevicius et al., 2018). Prior work shows that bfloat16, which consists of $8$ mantissa bits and $8$ exponent bits, is capable of training DNN models with FP32 accuracy (Wang & Kanwar, 2019). In the light of this, we hypothesize that HBFP6 with 6 mantissa bits and 10 exponent bits can also be used to train DNN models. Our experiments show that HBFP6 with a block size of $1$ is capable of training models with FP32 accuracy. As a matter of fact, HBFP6 can train DNN models with block sizes up to $256$.

## 3.2 FURTHER MINIMIZING HBFP

Block floating point shares a single exponent across a block of values using the exponent of the largest tensor value. Since block floating point format does not apply normalization ($2^{exponent} \times 0.mantissa$ instead of $2^{exponent} \times 1.mantissa$), the precision within a block is highly dependent on the largest element in that block, which decides on the exponent value. The interval between two consecutive representable numbers is calculated as in Equation 1. As the number of elements sharing the same exponent (block size) increases, variance of the values in a block significantly increases, which leads to precision loss for the small elements in the block. As the number of mantissa bits decreases, the model's sensitivity to the block size increases because the interval increases leading to higher quantization errors. Larger mantissa bit widths make the distribution more resilient to the quantization error and to larger block sizes, as each tensor element can be represented more accurately.

$$interval = \frac{2^{largest\ exponent}}{2^{\#\ of\ mantissa\ bits}} \tag{1}$$

As prior work (Drumond et al., 2018; Drumond, 2020) only investigated power-of-two mantissa bits and focused mostly on the design space of HBFP8, the interplay between the number of mantissa bits and the block size is left unexplored. We investigate the whole design space of HBFP varying both parameters, and claim that reducing the block size will enable reducing the number of mantissa bits, and thus improve hardware efficiency. As we have shown in Section 2, larger block sizes are more desirable for hardware efficiency. To minimize HBFP one step further, we investigate existing techniques to recover the data loss due to quantization.

For CNN models, prior work indicates that the first convolution layer and the last fully connected layer have a larger impact on the final model accuracy and keeping these layers in FP32 allows for reducing precision in the rest of the layers (Choi et al., 2018; Zhou et al., 2018; Mellempudi et al., 2019; Wang et al., 2018). The first layer takes the input images and filters the images with several convolution kernels and returns feature maps. Thus, it is critical for the final model to keep the input information fully accurate and to preserve the data in the initial feature map. The last layer of a CNN returns a probability distribution over the possible outcomes for image classification which makes it crucial to retain information better for this layer.

According to our calculations, the first and last layers of the CNN models account for a negligible amount of computation; and thus, keeping them in high precision during HBFP training does not result in a significant increase in the hardware area or energy consumption. For one of the smallest CNN models ResNet20, these layers take up only $1.08\%$ of the total number of FLOPs in the model.

For larger models, the percentage is even lower (e.g., $0.39\%$ for ResNet50, $0.27\%$ for ResNet74). Thus keeping these layers in FP32 helps further minimize HBFP while not introducing a significant hardware overhead.

### 3.3 ACCURACY BOOSTERS

In this section, we show how we can leverage mixed-precision training for minimizing HBFP without sacrificing accuracy. Existing mixed-precision techniques include varying the precision between different layers, functions, and epochs. FracTrain gradually increases the precision of weights, activations, and gradients during training (Fu et al., 2020). It also introduces an algorithm adapting the precision of activations and gradients in different layers depending on the input. Fu et al. (2021) argue that low precision due to quantization affects DNN training similarly to exploration with a high learning rate and introduce CPT, a periodic precision schedule for low-precision DNN training. In both of these methods and in contrast to our work, the backward pass uses a higher precision than the forward pass ($6 - 8$ vs. $3 - 8$ bits). The backward pass usually takes around $3\times$ as long as the forward pass; therefore, CPT spends most of the time in 8- and 6-bit fixed-point computations. Moreover, the silicon area and energy gain that these fixed-point calculations provide over FP32 are $9\times$ and $15\times$ respectively. In comparison, HBFP6 and HBFP4 provide $10\times$ and $17\times$ area and power gain respectively as we have computed in Section 2.

We leverage the insight that each training epoch has a different effect on the final model's accuracy. Rahaman et al. (2019) and Xu et al. (2019) show that DNNs first learn low-frequency components, where the frequency is defined for the coordinates of the input space. Xu et al. (2019) also empirically show that for CNN models, the high-frequency components have higher complexities and are learned in the last epochs. In light of these findings, we hypothesize that high-frequency functional components are more sensitive to quantization errors. Thus, higher precision is required for the last stage of DNN training, where the optimization occurs after an appropriate amount of generalization in the network. After reaching a certain loss value in our 4-bit training, switching the tensors back to 6-bit mantissas enable the sensitive fine-tuning performed in the final epochs and help increase the accuracy even more.

For most DNN models, using HBFP6 for the last epoch is sufficient to boost the final model accuracy, while the rest of the epochs are trained using HBFP4. We estimate the gain in silicon provisioning and power consumption by using the average of the individual gains of the numerical formats weighted with the fraction of the training time the numerical format is used. When we consider the number of epochs that each hardware is used for, the average area and power gain are calculated to be $1.5\times$ more than HBFP6 ($16.98\times$ over FP32).

## 4 EXPERIMENTAL RESULTS

In this section, we will first describe our experimental setup, then we will explain and discuss our experimental results on the state-of-the-art models and datasets for various DNN tasks. We trained ResNet20/50/74 (He et al., 2016), and DenseNet40 (Huang et al., 2016) on CIFAR10 and CIFAR100 (Krizhevsky & Hinton, 2009) datasets for image classification. We also trained a Transformer-Base (Vaswani et al., 2017) on the WMT16 English-German dataset for machine translation. We use FP32 as baseline for both model accuracies and hardware comparisons. To evaluate the impact of our method, we tune the models using FP32, and then train the same models from scratch with the same hyperparameters in HBFP. For the image classification experiments, we report Top1 validation accuracies; for machine translation, we report BLEU scores.

### 4.1 MINIMIZING HBFP

Table 1 shows the Top1 validation accuracies for ResNet20, ResNet50, ResNet74, and DenseNet40 on CIFAR10 and CIFAR100 datasets trained with various HBFP configurations. We observe that HBFP6 is the smallest HBFP configuration that gives accuracies within $2\%$ of FP32 accuracy for block sizes up to 256. We also observe that unnecessarily large block sizes like 576 (providing only $0.8\%$ silicon area gain compared to a block size of 256) hurt accuracy for some models. Larger blocks will contain a larger variety of values in terms of magnitude (affected e.g. by outliers), so will result in larger approximation errors than smaller blocks and lower accuracy in training.

Table 1: Top-1 validation accuracies of various CNN models for various HBFP configurations

| Number Format | Block Size / Area Gain | Models and Datasets | | | | |
|---|---|---|---|---|---|---|
| | | CIFAR10 | | CIFAR100 | | |
| | | ResNet20 | ResNet74 | ResNet50 | ResNet74 | DenseNet40 |
| **FP32** | - | 91.72 | 93.57 | 74.11 | 74.55 | 72.42 |
| **HBFP8** | **576** / 7.23 | 91.52 | 93.36 | 73.76 | 74.32 | 73.73 |
| HBFP6 | **16** / 8.85 | 91.12 | 93.38 | 73.11 | 73.51 | 72.08 |
| | **25** / 9.29 | 91.09 | 92.54 | 72.59 | 73.20 | 71.77 |
| | **36** / 9.71 | 91.29 | 92.61 | 72.45 | 72.87 | 71.83 |
| | **49** / 10.00 | 91.33 | 92.93 | 72.34 | 72.40 | 71.87 |
| | **64** / 10.19 | 91.12 | 92.93 | 72.12 | 72.40 | 71.81 |
| | **256** / 10.71 | 91.38 | 92.79 | 72.01 | 72.53 | 71.50 |
| | **576** / 10.81 | 90.65 | 92.19 | 70.80 | 72.51 | 71.02 |
| HBFP5 | **16** / 10.02 | 90.18 | 92.24 | 67.15 | 70.52 | 69.36 |
| | **25** / 10.95 | 90.21 | 92.04 | 67.85 | 69.03 | 70.00 |
| | **36** / 11.55 | 90.12 | 91.99 | 67.12 | 69.91 | 69.01 |
| | **49** / 11.94 | 89.68 | 91.96 | 68.72 | 68.67 | 69.33 |
| | **64** / 12.21 | 89.72 | 91.32 | 66.81 | 69.29 | 68.85 |
| | **256** / 12.92 | 89.13 | 91.13 | 63.01 | 66.57 | 67.84 |
| | **576** / 13.07 | 88.91 | 90.98 | 64.61 | 64.41 | 67.57 |
| HBFP5 with first and last layers in FP32 | **16** / $-$ | 90.80 | 92.56 | 71.23 | 72.31 | 70.90 |
| | **25** / $-$ | 90.62 | 91.79 | 71.76 | 72.58 | 70.52 |
| | **36** / $-$ | 90.62 | 92.73 | 71.45 | 72.26 | 70.01 |
| | **49** / $-$ | 90.49 | 92.02 | 70.51 | 72.07 | 70.16 |
| | **64** / $-$ | 90.31 | 92.41 | 69.59 | 72.09 | 70.10 |
| | **256** / $-$ | 90.07 | 92.11 | 68.76 | 70.20 | 69.21 |
| | **576** / $-$ | 90.37 | 91.06 | 68.66 | 69.14 | 68.78 |
| HBFP4 | **16** / 12.52 | 82.59 | 76.85 | - | - | 63.70 |
| | **25** / 14.03 | 81.82 | 78.62 | - | - | 64.25 |
| | **36** / 15.01 | 80.84 | 76.64 | - | - | 63.34 |
| | **49** / 15.68 | 79.32 | 71.19 | - | - | 65.55 |
| | **64** / 16.15 | 80.18 | 74.35 | - | - | 62.37 |
| | **256** / 17.43 | 76.96 | 60.65 | - | - | 60.02 |
| | **576** / 17.69 | 75.33 | 66.70 | - | - | 59.77 |
| **Total Number of FLOPs required to train the model** | | 41M | 174M | 119M | 326M | 542M |

Table 1 also shows that for the smallest model ResNet20, decreasing the block size of HBFP5 to 25 leads to $1.5\%$ accuracy degradation but $1.1\times$ hardware gain compared to HBFP6 in return (see Figure 1). As the block size increases, the accuracy decreases more for HBFP5. Among larger models, ResNet74 trained on CIFAR10 gives similar results to ResNet20 on CIFAR10 ($1.3\%$ accuracy degradation with HBFP5). ResNet50 and ResNet74 on CIFAR100 do not perform well with HBFP5 unlike smaller variations, and result in up to $7\%$ accuracy degradation. HBFP5 incurs $3 - 4\%$ accuracy degradation on average for moderate block sizes (e.g., 49, 64). We show that for almost all DNN models, keeping the first and last layers in FP32 enables HBFP5 training. For ResNet20 and ResNet74 on CIFAR10, the accuracy drop is around $1\%$ relative to FP32 accuracy for block size of 64 while bringing $1.22\times$ silicon area and energy gain. ResNet50 and ResNet74 on CIFAR100 accuracies improve significantly compared to naive HBFP5 for moderate block sizes (e.g., 49, 64), and the models get more sensitive as the block size increases to 256 and higher.

We also report HBFP4 accuracies to show the limitations of HBFP format. Even for the small models like ResNet20, with block size of 16, the accuracy drops more then $9\%$. As the accuracy drop for ResNet50 and ResNet74 on CIFAR100 is considerably high even with HBFP(5, 16), we

did not train these models with HBFP4. We observe that for HBFP4, the sensitivity to the block size increases for all the models because the distortions in the tensor distributions increase. So, block sizes larger than 16 incur even more dramatic accuracy drops like 26.87% for ResNet74, showing the significance of only one bit of mantissa for the accuracy.

## 4.2 ACCURACY BOOSTERS

Considering the observed accuracy values in Table 1 and our hardware model explained in Section 2, a block size of 49 is within 95% of the maximum hardware gain (the point where the HBFP6 line in Figure 1 levels off) while achieving accuracies with less than 1% degradation. Thus, we choose block size of 49 as the sweet spot and perform Accuracy Boosters using this block size. We perform the last epoch of the training in HBFP$(6, 49)$ and the rest in HBFP$(4, 49)$ for all the experimental settings. We also trained by keeping the last 10 epochs in HBFP$(6, 49)$ to observe the improvement in accuracy for the CNN models. We can see that for most of the CNN models, our proposed technique outperforms FP32. When we keep the last 10 epochs in HBFP6, we observe that the accuracies slightly increase. We also trained with block size of 256 by using Accuracy Boosters and observe that the accuracy loss is less than 1% for most of the models and DenseNet40 still outperforms FP32 accuracy(see Table 2).

Table 2: Top-1 validation accuracies of various CNN models for Accuracy Boosters

| | | Models and Datasets | | | | |
| | | CIFAR10 | | CIFAR100 | | |
| **Epochs using HBFP6** | **Block Size** | ResNet20 | ResNet74 | ResNet50 | ResNet74 | DenseNet40 |
|---|---|---|---|---|---|---|
| Only last | 49 | 91.45 | 92.85 | **74.25** | 74.39 | **74.06** |
| Only last | 256 | 90.76 | 92.40 | 72.78 | 74.32 | **74.77** |
| Last 10 | 49 | 91.69 | 92.87 | **74.50** | **74.45** | **74.26** |
| **FP32** | - | 91.72 | 93.57 | 74.11 | 74.55 | 72.42 |

For Transformer, Accuracy Boosters achieves a BLEU Score of 25.08 when used only for the last epoch and 25.40 when used for the last 10 epochs, while the FP32 score is 26.09. We also observe that for Transformer, while HBFP6 outperforms FP32, HBFP4 does not incur high amounts of accuracy loss but Accuracy Boosters still increase the accuracy to the level close to FP32 (see Table 3). The total number of calculations required to train Transformer-Base is $3.3 \times 10^{18}$ FLOPs.

Table 3: BLEU Scores for Transformer-Base trained on WMT16
English-German dataset with various training techniques

| | **Training Technique** | | | | |
| | FP32 | HBFP$(6, 49)$ | HBFP$(4, 49)$ | booster (last) | booster (last 5) |
|---|---|---|---|---|---|
| **BLEU Score** | 26.09 | 26.16 | 24.73 | 25.08 | 25.40 |

The Accuracy Boosters method offers up to $16.98\times$ hardware gain compared to FP32, using only 4 bits for 99.7% of the total training time, while having a comparable or better accuracy. Table 2 shows the Top1 validation accuracies for various models with Accuracy Boosters at last 1 and 10 epochs. Models trained on CIFAR10 are trained for 160 epochs, whereas for CIFAR100, the total number of epochs for all models is 300. The transformer is trained for 35 epochs and Accuracy Boosters are applied in the last 1 and 5 epochs. Figure 2 illustrates the training process of various CNN models and datasets trained with FP32, HBFP$(6, 49)$, and Accuracy Boosters.

## 5 RELATED WORK

In recent years, there has been a significant amount of research on inference and training (Courbariaux et al., 2015a; Hubara et al., 2016; Rastegari et al., 2016; Zhou et al., 2018; Li & Liu, 2016;

Table 4: Power consumption ratios between FP32 and various Accuracy Booster configurations.

| | **Model** | | | | | |
|---|---|---|---|---|---|---|
| | ResNet20 CIFAR10 | ResNet74 CIFAR10 | ResNet50 CIFAR100 | ResNet74 CIFAR100 | Transformer (last) | Transformer (last 5) |
| **Ratio** | 28.2 | 36.2 | 35.0 | 36.5 | 38.0 | 30.0 |

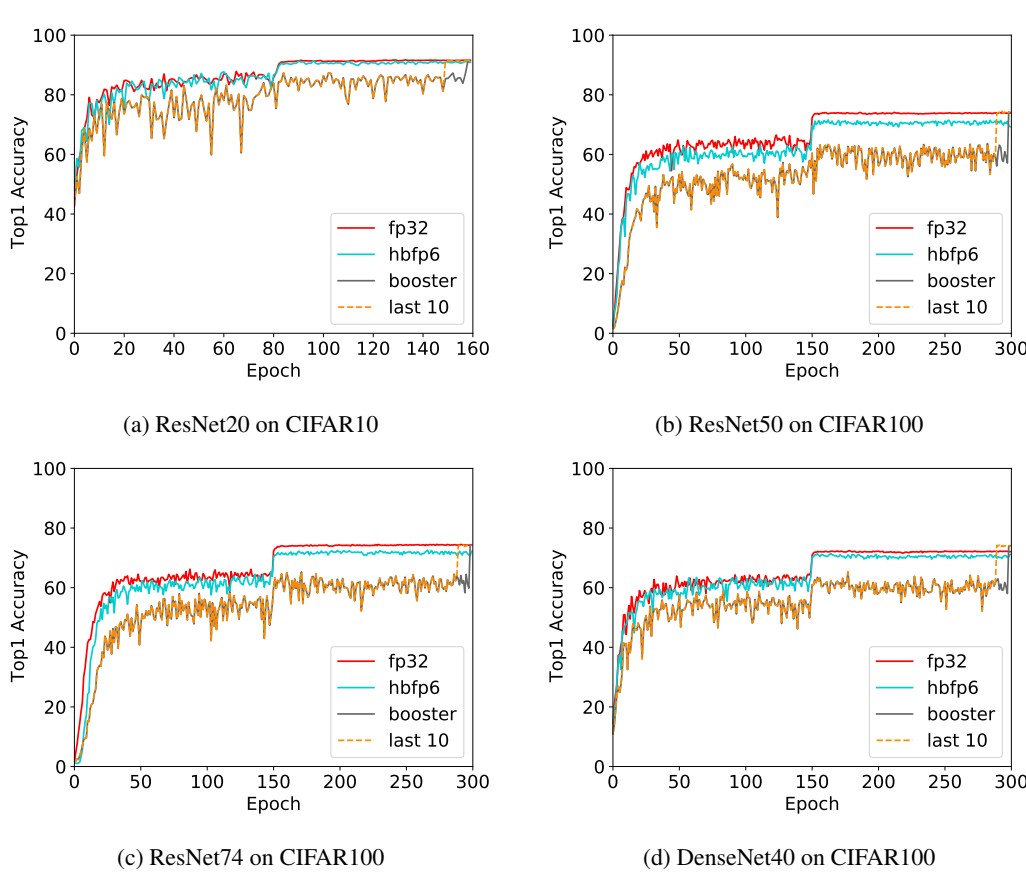

(a) ResNet20 on CIFAR10

(b) ResNet50 on CIFAR100

(c) ResNet74 on CIFAR100

(d) DenseNet40 on CIFAR100

Figure 2: Top1 Accuracies for various models with various training techniques.

Lin et al., 2016; Courbariaux et al., 2015b; Dettmers et al., 2022a) with narrow numerical representations. Google Brain has introduced bfloat16 as a drop-in replacement for FP32 (the IEEE 754 single-precision floating-point format) in accelerators, using the same number of exponent bits as FP32 with half the number of overall bits (Wang & Kanwar, 2019). NVIDIA's mixed-precision training proposes to use FP16 (IEEE 754 half-precision) for training and FP32 for data loss recovery (e.g., for weight updates) (Micikevicius et al., 2018). Prior work proposed training models with FP8, either using mixed precision training or customizing the number of bits provisioned for the exponent and mantissa (Sun et al., 2019). The provisioning is done according to the precision requirement difference in the forward and backward passes. Recent research advocates the use of Block Floating-Point for DNN training (Drumond, 2020) and inference (Darvish Rouhani et al., 2020). Flexpoint (Köster et al., 2017) and Dynamic Fixed-Point (Das et al., 2018) propose block-floating-point formats for training with a 16-bit mantissa and a shared exponent. Prior work proposed a novel format for training DNNs with BFP, called Hybrid Block Floating-Point (HBFP) (Drumond et al., 2018). In this paper, we argue that reducing the mantissa bit width in HBFP results in a significant improvement in silicon efficiency while designing hardware for DNN training.

Many have proposed techniques to compensate for the data loss introduced by narrower numerical representations. Mellempudi et al. (2019) use mixed precision with enhanced loss scaling to train models accurately with FP8. Sun et al. (2019) leverage the precision requirement difference in the forward and backward passes and propose a hybrid FP8 format where they vary the number of bits provisioned for the exponent/mantissa. Sun et al. (2020) show that mantissa-free radix-4 FP4 training is possible with adaptive gradient scaling and two-phase rounding.

Mixed-precision training has emerged as a popular technique to increase the fraction of leaner arithmetic formats within the training process, motivating us to explore the design space of mixed-mantissa HBFP. Several techniques vary the precision layer-wise by using higher precision arithmetic for layers with greater significance (Khoram & Li, 2018; Yang & Jin, 2021; Shen et al., 2020). Specifically, Choi et al. (2018); Zhou et al. (2018); Mellempudi et al. (2019); Wang et al. (2018) use FP32 for the first and last layers. Fu et al. (2021) employ fixed-point arithmetic with different bit widths epoch-wise over the course of training. Combining the layer-wise and epoch-wise approaches, Fu et al. (2020); Zhang et al. (2021); Noh et al. (2022) vary the precision adaptively per epoch and layer at the same time using control mechanisms. While all the aforementioned studies employ leaner arithmetic for a fraction of the training process, they fail to make the leaner arithmetic the common case of the training process. In particular, Fu et al. (2021) and Fu et al. (2020) use high-precision arithmetic (8-bit or 6-bit) for the backward pass, which is more memory-bandwidth intensive and requires up to $3\times$ more computation than the forward pass.

Recent work (Dettmers et al., 2022b) suggests that during mixed-precision FP16 training, the optimizer states can be reduced to 8 bits by using a block-wise quantization method. This observation is in line with our work that applies quantization by extracting the largest exponent per block. Similarly, FAST (Zhang et al., 2021) uses a block-floating-point-based layer-wise mixed precision approach using 2 and 4-bit mantissas. Unlike our work, FAST requires fine-tuning several additional hyperparameters for its training algorithm, making it difficult to apply to other DNN models. Another block-floating-point-based work, FlexBlock (Noh et al., 2022), uses 4 and 8-bit mantissas with various block sizes and also needs higher-precision block-floating-point formats only for weight gradient calculations that suffer more from quantization errors.

## 6 CONCLUSION

The immense growth in DNN model sizes and complexities, and exploding amount of data have led to a commensurate increase in demand for computing, and thus the design of silicon-efficient DNN accelerators. Low-precision training techniques and specialized numerical formats have been introduced to increase the arithmetic density of the DNN accelerators. One of such formats, Hybrid Block Floating-Point (HBFP), that allows for a majority of the DNN's arithmetic operations (i.e., dot products) to be performed using fixed-point arithmetic and limits floating-point arithmetic to a minimal set of operations, has been shown to achieve FP32 accuracy with 8-bit mantissas. Moreover, lower number of mantissa bits allows for exceptional improvements in hardware (e.g., up to $17.5\times$ gain over FP32 silicon area). In this paper, we investigate the empirical limits of HBFP for emerging models and datasets. We show that HBFP6 is the smallest HBFP format achieving FP32 accuracy. To further minimize the number of bits, we analyze the effect of the block size and show that for small models, lowering the block size to 25 allows for training with HBFP5. Keeping the first and last layers in FP32 enables HBFP5 training with block sizes of 49 and higher for all models. We propose the Accuracy Boosters technique to bring HBFP4 into training, varying the number of mantissa bits between 4 and 6 throughout epochs, leveraging the insight that each epoch has a different effect on training. We show that the last stage of training requires more precision than the rest. Our method achieves a $16.98\times$ hardware overhead gain over FP32, while outperforming FP32 accuracy.

## 7 REPRODUCIBILITY STATEMENT

All the experimental results reported in the paper are reproducible using the code submitted as a supplementary material. The models, datasets and hyperparameters used in the training of all the models are explained in Section 4, Appendix A and B. Both image classification and machine translation tasks have scripts to train the models mentioned in the paper.

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

## A    HYPERPARAMETERS FOR IMAGE CLASSIFICATION EXPERIMENTS

We report the hyperparameters we use for all our experiments in Table 5. All convolutional layer weights are initialized using a zero-mean Gaussian distribution whose standard deviation (std) is $\sqrt{2n_l}$ ($n_l$ is the number of activations in the output), a method introduced in He et al. (2015). All the Batch Normalization layer weights are initialized to 1 and are kept in FP32 as HBFP suggests (Drumond et al., 2018).

Table 5: Hyperparameters for various sizes of ResNet on CIFAR10 and CIFAR100

| Hyperparameter | Resnet20 & ResNet74 on CIFAR10 | Resnet50 & ResNet74 on CIFAR100 |
|---|---|---|
| Epochs | 160 | 300 |
| Batch size | 128 | 128 |
| Optimizer | SGD | SGD |
| Nesterov | True | True |
| Learning rate | 0.1 | 0.1 |
| Learning rate decay epochs | 82, 122 | 150, 225 |
| Learning rate warmup | True | True |
| Weight decay | 1e-04 | 1e-04 |
| Momentum | 0.9 | 0.9 |

## B    HYPERPARAMETERS FOR NEURAL MACHINE TRANSLATION EXPERIMENTS

Our implementation is based on Transformer implementation of Fairseq (Ott et al., 2019). We report the hyperparameters we use for all our experiments in Table 6. For the ones we do not specify, we use the default hyperparameters.

Table 6: Hyperparameters for Transformer-Base on WMT'16 Translation Task English to German

| Transformer architecture hyperparameters | |
|---|---|
| Dataset | wmt16_en_de_bpe32k |
| Architecture | transformer_wmt_en_de |
| Number of layers | 6 |
| Number of heads | 8 |
| Number of hidden dimensions | 512 |
| Optimizer | Adam |
| Adam-betas | $(0.9, 0.98)$ |
| Learning rate | 0.0007 |
| Learning rate scheduler | Inverse Square Root |
| Warmup updates | 4000 |
| Warmup initial learning rate | 1e-07 |
| Label smoothing | 0.1 |
| Maximum tokens | 3584 |
| Criterion | label_smoothed_cross_entropy |
| Clip-norm | 0.0 |
| Weight decay | 0.0 |
| Dropout | 0.1 |
| Update frequency | 2 |
| Share all embeddings | True |

