# OpenReview forum: "Accuracy Boosters: Epoch-Driven Mixed-Mantissa Block Floating-Point for DNN Training"
_ICLR.cc/2023/Conference — Submitted to ICLR 2023_

### Official Review · Reviewer_gycw · 2022-10-17

**Confidence:** 4
**Correctness:** 3
**Technical Novelty And Significance:** 3
**Empirical Novelty And Significance:** 2
**Recommendation:** 6

**Clarity, Quality, Novelty And Reproducibility:**

The paper is well written and the work is well explained.
The paper presents original incremental work upon previous research on HBFP.
Code was provided and the experiments can be reproduced.
Some improvements on figure2, could make the papers better: figure2 should highlight the last 10 epochs to show accuracy benefit from accuracy booster.
Table 1 should add flop to train fp32 for ease of comparison.


**Details Of Ethics Concerns:**

No ethics issues

**Strength And Weaknesses:**

The paper is well written and clear. The paper shows comparison between various HBFP configurations and presents accuracy booster method


Few points that could improve the paper:

It would be interesting to see trade-off accuracy and hardware efficiency when the variable block size is across different layers.

The paper should also evaluate on bigger dataset: Imagenet and updated model: ViT

Table 1: should have HBFP8 for comparison

HBFP6 with block size 1, is same as having 16bit for every value, wouldn't FP16 with mixed-precision training be better or equivalent in terms of accuracy?

In table1, HBFP6 doesnt achieve same accuracy of FP32 model. Thus, the intro's claim should be revisited: "We show that HBFP6 is the smallest HBFP format achieving FP32 accuracy".

In figure2, should add some form to show power savings while using accuracy booster.


**Summary Of The Paper:**

The paper investigates the HBFP parameter search space to further improve efficiency and density in hardware accelerators.
They propose training on HBFP6, after experiments making the trade off: lower block size gives more accuracy, but is less HW efficient.
They show accuracy booster method that switch number of mantissa bits in different epochs in the training stage. They evaluate their experiments on CNN and transformers for cifar10/100 and wmt16.

**Summary Of The Review:**

Overall the paper is well written and clear to understand.
The paper should show additional data or experiments that will make the claims and proposed methods even stronger.

---

> ### Author Response · Authors · 2022-11-15
> **Addressing your concerns and suggestions**
>
> Thank you very much for your valuable comments and suggestions to improve the paper. We address each of your concerns below and are happy to discuss further.
>
> > It would be interesting to see tradeoff accuracy and hardware efficiency when the variable block size is across different layers.
>
> We agree that it is an interesting idea to vary the block size between the layers, and it is orthogonal to Accuracy Boosters, so even a combination of these techniques could be useful to minimize HBFP even further. We definitely plan to include that direction in our future work. However, for this rebuttal, we unfortunately don't have enough time to investigate this thoroughly.
>
> > The paper should also evaluate on bigger dataset: Imagenet and updated model: ViT
>
> We are running further experiments on Transformers and ImageNet, and we are planning to provide the results by the end of the rebuttal period as they are taking a long time to finish.
>
> > Table 1: should have HBFP8 for comparison
>
> Thank you for this suggestion. We updated the paper accordingly.
>
> > HBFP6 with block size 1 is the same as having 16bit for every value, wouldn't FP16 with mixed-precision training be better or equivalent in terms of accuracy?
>
> FP16 requires mixed precision (with FP32) to reach FP32 accuracy. As we stated in our paper, FP16 without mixed-precision training does not achieve enough accuracy (Micikevicius et al., 2018). However, HBFP6 by itself (without FP32 arithmetic) can train DNN models with block sizes starting from 1 up to 256, which would provide considerable hardware overhead improvement.
>
> > In table1, HBFP6 doesnt achieve same accuracy of FP32 model. Thus, the intro's claim should be revisited: "We show that HBFP6 is the smallest HBFP format achieving FP32 accuracy".
>
> We took a 1% accuracy difference as a threshold for FP32 accuracy, and we included that claim. However, we agree that it might lead to misunderstanding. Thus, we changed the sentence to: "... achieving competitive accuracies".
>
> > In figure2, should add some form to show power savings while using accuracy booster.
>
> Following your suggestion, we included a new table with the power savings of various HBFP configurations (Table 3). The numbers show the ratio between the power consumption of FP32 hardware and various HBFP configurations for various models.
>
> > Some improvements on figure2, could make the papers better: figure2 should highlight the last 10 epochs to show accuracy benefit from accuracy booster.
>
> We updated the figure accordingly. We removed the ResNet74 on CIFAR10 plot because of the page limitation. However, the results for that model are still included in Table 2.
>
> > Table 1 should add flop to train fp32 for ease of comparison.
>
> The FLOP numbers in Table 1 are for FP32, and they are given to inform the reader about the number of floating point operations needed to train each model. With HBFP, approximately the same number of operations will be performed in fixed point arithmetic instead, except in the cases of HBFP5 with first/last layers in FP32.

---

### Official Review · Reviewer_VDoK · 2022-10-21

**Confidence:** 5
**Correctness:** 3
**Technical Novelty And Significance:** 1
**Empirical Novelty And Significance:** 2
**Recommendation:** 3

**Clarity, Quality, Novelty And Reproducibility:**

Questions:
1. Which axis of the CNN tensors is used for exponent sharing? I assume it's input channels but I would like a confirmation.
2. Why was a 10-bit exponent used for HBFP? Why not an 8-bit exponent like FP32 and bfloat16? This would make it easier for both software and hardware to operate between these different formats.
3. Why does using unnecessarily large block size hurt the model accuracy? I assume tensors would be padded with zeros to meet the block size requirement, which shouldn't affect accuracy at all.
4. Why use non-power of two block sizes like 49? A power of two block size makes sense since doing matrix multiplies usually requires an adder tree in hardware. This tree is naturally power of two sized.

Table 3 should say "Epochs using HBFP6".

**Strength And Weaknesses:**

Strengths:
1. Paper was easy to understand
2. HBFP seems like a interesting data format with potential

Weaknesses:
1. The paper is entirely empirical. All conclusions are drawn largely from experiments on CIFAR-10/100 using ResNets and DenseNet40. One 1 table had data on transformers. This is a very weak set of benchmarks and models for an empirical paper.
2. The results are not too exciting. For CNNs, the first and last layers must be kept in FP32 - so HBFP is only used for the "easy to quantize" layers. For the Accuracy Booster technique, it doesn't seem to work well on transformer.
3. The paper combines two existing techniques: HBFP and FracTrain. The techniques are mostly orthogonal and not combined in a novel or insightful way. I didn't get much out of the paper besides "certain configurations of HBFP4/5/6 can be used to train CNNs", which isn't all that surprising.

**Summary Of The Paper:**

The authors empirically study hybrid block floating-point (HBFP). HBFP uses a single shared exponent with a block of multiple mantissas. The authors run training experiments on CIFAR-10/100 using ResNets and DeseNet40 with HBFP. Mantissa bits of 6, 5, and 4 are tested with block size ranging from 16 to 576. Both forward and backward pass is quantized to HBFP. The authors conclude that HBFP5 with the first and last layers in FP32 is sufficient to match FP32 training accuracy.

The authors also experiment with FracTrain, where most epochs of training is done in low precision and the final one or few epochs in higher precision. For CNNs, training in HBFP4 with last epoch in HBFP6 recovers FP32 accuracy. However, for Transformer there was still a gap between this setup and FP32.

**Summary Of The Review:**

A purely empirical paper that studies the HBFP format for DNN training. The paper does not have much novelty and the experimental section is weak, with data mostly coming from ResNets trained on CIFAR-10/100. The main takeaway is that training with HBFP5 works fine for CNNs. If training with HBFP4, the last few epochs should be done in higher precision like HBFP6.

---

> ### Author Response · Authors · 2022-11-15
> **Addressing your concerns about the novelty of our work and our experiments**
>
> Thank you very much for your detailed comments and questions. Please find below our answers to each of your questions and concerns. We are happy to discuss this further.
>
> > The paper combines two existing techniques: HBFP and FracTrain. The techniques are mostly orthogonal and not combined in a novel or insightful way. I didn't get much out of the paper besides "certain configurations of HBFP4/5/6 can be used to train CNNs", which isn't all that surprising.
> >  The authors also experiment with FracTrain, where most epochs of training is done in low precision and the final one or few epochs in higher precision.
>
> There is a misunderstanding that we combine HBFP and FracTrain, while Accuracy Boosters has the following differences from FracTrain:
>
>
> 1. FracTrain uses different precisions for forward and backward passes. It uses higher precision for backward pass, which contains around 75% of the calculations. Accuracy Boosters uses the same precision for forward and backward passes.
>
> 2. FracTrain divides the training into four stages and gradually increases the precision (from 3 to 8 for forward, from 6 to 8 for backward passes). Accuracy Boosters does not go beyond 6-bit arithmetic and uses high precision only at the last epoch. So almost all of the training is done with 4-bit arithmetic (We also show that if we use high precision for the last ten epochs we outperform FP32)
>
> > The paper is entirely empirical.
> > The results are not too exciting. For CNNs, the first and last layers must be kept in FP32 - so HBFP is only used for the "easy to quantize" layers. For the Accuracy Booster technique, it doesn't seem to work well on transformer.
>
> We updated the second and the last paragraph of the introduction highlighting the novelty of our work, and carried the paragraph about prior work in the introduction (which causes confusion/misunderstandings about our contributions) to the related work section.
>
> To the best of our knowledge, there is no prior work proving a full investigation of the parameter space of block floating point. We do not claim that using full precision first and last layers is our contribution. It is a method that we included in our study to explore the potential of HBFP fully.
>
> Accuracy Boosters is the first method that introduces high precision in the training process only for the last epoch and uses low precision for 99.7% of the training. Prior work using epoch- and/or layer-wise mixed-precision fails to make leaner arithmetic the common case for the training process. Our work shows, for the first time, that training DNNs with 4-bit arithmetic is possible with a last booster/fine-tuning touch.
>
> We are running further experiments on Transformers and ImageNet, and we are planning to provide the results by the end of the rebuttal period as they are taking a long time to finish.

---

> > ### Author Response · Authors · 2022-11-15
> > **Answers to the listed questions**
> >
> >
> > > Which axis of the CNN tensors is used for exponent sharing? I assume it's input channels but I would like a confirmation.
> >
> > We flatten the tensors to place the values stored in contiguous memory locations in the same block. While forming the blocks, the H and W dimensions of the CNN tensors are used, and exponents are not shared among different input channels ( C ).
> >
> > > Why was a 10-bit exponent used for HBFP? Why not an 8-bit exponent like FP32 and bfloat16? This would make it easier for both software and hardware to operate between these different formats.
> >
> > HBFP (Drumond et al., 2018) uses 10-bit exponents stating that the size of the exponent has no effect on the hardware. HBFP implementation is public ([https://github.com/parsa-epfl/HBFPEmulator](https://github.com/parsa-epfl/HBFPEmulator)) and we started to conduct our research with the HBFPEmulator code. However, after the submission, we also tested with 8-bit exponents to fully explore the design space and observed that we obtain the same accuracies with 8 bits of exponent.
> >
> > > Why does using unnecessarily large block size hurt the model accuracy? I assume tensors would be padded with zeros to meet the block size requirement, which shouldn't affect accuracy at all.
> >
> > Thank you very much for your valuable question, which led us to make the explanation for large block sizes more precise. We updated the paper accordingly. Larger blocks will contain a larger variety of values in terms of magnitude (affected e.g. by outliers), so it will result in larger approximation errors than smaller blocks and lower accuracy in training.
> >
> > > Why use non-power of two block sizes like 49? A power of two block size makes sense since doing matrix multiplies usually requires an adder tree in hardware. This tree is naturally power of two sized.
> >
> > We chose the number 49 as a sweet spot between hardware improvement and model accuracy. Our further experiments show that one could apply Accuracy Boosters with up to block sizes of 256 (see the second row of Table 2). We agree that power-of-two block sizes will give more benefits in the hardware perspective, thus one can definitely use block sizes of 64 or 256 and still achieve FP32 accuracy.
> >
> > >  Table 3 should say "Epochs using HBFP6".
> >
> > Thank you for pointing this typo out. We updated the table.

---

> > > ### Comment · Reviewer_VDoK · 2022-11-28
> > > **Major issue with exponent axes?**
> > >
> > > >We flatten the tensors to place the values stored in contiguous memory locations in the same block. While forming the blocks, the H and W dimensions of the CNN tensors are used, and exponents are not shared among different input channels ( C ).
> > >
> > > This seems to me like a major issue in the experiments. With block floating-point the whole point is to do make sure dot products can be computed using integer MACs (by factoring the shared exponents out). With convolutions the easiest way to do this is to share exponents along the channels for both inputs activations and weight filters. In your case it seems that exponents would be shared along W, but modern CNNs mostly use 1x1 and 3x3 convolutions, so we can only do a length-1 or length-3 integer dot product before encountering a different exponent. Can you describe how your HBFP layout would work efficiently in real hardware?

---

> > > > ### Author Response · Authors · 2022-11-30
> > > > **On the hardware aspect**
> > > >
> > > > When performing convolutional operations, there will be fold/unfold units that can batch multiple filters together while keeping their exponents separate. This can be achieved efficiently, thanks to the systolic array based design where we can feed exponents for each filter on a row with a slight modification to the exponent circuitry. This was the integer MACs can still be fully utilized.

---

> > > > > ### Comment · Reviewer_VDoK · 2022-11-30
> > > > > **Hardware aspect**
> > > > >
> > > > > You'll have to be more specific. Say I have a 3x3 convolution with input height and width 128, and the block size is 128. From my view if you simply unroll the spatial dimensions the activations will have a different shared exponent along each row. But the filter is only size 3 along the rows so you're limited to integer MAC of size 3 before having to convert to floating-point to deal with different exponents. If I use 1x1 convolution you are limited to size 1. How would you map these cases to hardware efficiently?

---

> > > > > > ### Author Response · Authors · 2022-12-12
> > > > > > **Hardware aspect**
> > > > > >
> > > > > > In this case, the exponents will be shared among the feature maps as well as height and width dimensions depending on the block size (when the block size is higher than 9). We agree that finding the optimal block dimension is a hard problem and we are still investigating it and we will include it in our future work. Currently, we are following the method in the original HBFP paper "Training DNNs with Hybrid Block Floating Point" by Drumond et al. Their code is open source under: https://github.com/parsa-epfl/HBFPEmulator and our method is built on top of it.

---

### Official Review · Reviewer_FLAx · 2022-10-25

**Confidence:** 2
**Correctness:** 4
**Technical Novelty And Significance:** 3
**Empirical Novelty And Significance:** 3
**Recommendation:** 8

**Clarity, Quality, Novelty And Reproducibility:**

Novelty: I am not an expert in fixed point formats. The method seems novel (HBFP mid precision training). However, there was no automatic technique to find how the best mixed precision training schedule.

Clarity: I found the paper was well written with a good background section.

**Strength And Weaknesses:**

Strengths:

* Good background section that clearly explains the use of HBFP
* Intuitive explanation of mixed precision training in the context of HBFP with proper experimental backing

Weaknesses:

* It is unclear why training at the very end using HBFP6 allows you to recover the accuracy loss, more explanation would have been better

**Summary Of The Paper:**

The paper introduces "accuracy boosters" that aim to train DNNs using HBFP4 for most of the epochs while switching to HBFP6 for the last few epochs. Thy show that mixed precision training is applicable under HBFP setting and achieve similar accuracy with better hardware utilization.


**Summary Of The Review:**

I enjoyed reading the paper. I am not an expert in different fixed point formats. However, the background section was fairly detailed for me to appreciate the use of HBFP.

The technique seems fairly intuitive. The authors adapt mixed precision training in the context of HBFP. The results show that by training most of the time with HBFP4 and switching to HBFP6 at the end allows the authors to achieve comparable accuracy at a lower hardware cost.

That said the technique for finding this schedule was mainly through manual experimentation. Therefore, I am not sure of the novelty of the work.

---

> ### Author Response · Authors · 2022-11-15
> **About novelty of our work and our further insights**
>
> Thank you very much for your valuable comments and for recognizing the potential impact of our work. We address your concerns below and are happy to discuss further.
> > -   It is unclear why training at the very end using HBFP6 allows you to recover the accuracy loss, more explanation would have been better
>
> -   Our main insight in Accuracy Boosters is that HBFP4 has enough precision to bring the model in the correct direction for optimization. However, it lacks the precision to fine-tune the model. HBFP6 during the short booster period here plays the role of fine-tuning and boosting the accuracy to the FP32 level.
>
> -   Moreover, we argue that our results are supported by the frequency principle (Luo et al., 2019; Rahaman et al., 2019; Xu et al., 2019), which states that DNNs learn the low-frequency and low-complexity features earlier than the high-frequency and high-complexity features. Thus, we claim that HBFP4 has enough precision to learn low-frequency and low-complexity features; however, we need HBFP6 for the high-complexity features.
>
> > There was no automatic technique to find how the best mixed precision training schedule. The technique for finding the training schedule was mainly through manual experimentation. Therefore, I am not sure of the novelty of the work.
>
> The main idea of Accuracy Boosters is to enable DNN training with 4-bit arithmetic and obtain FP32 accuracy in the end. Therefore we are training almost all the model with HBFP4, and we bring HBFP6 only at the last epoch, which is already enough for accurate training. We additionally show that if we use HBFP6 for the last ten epochs we can even outperform FP32 training. Thus, our method does not require an automatic decision feature.

---

### Official Review · Reviewer_gLsp · 2022-10-28

**Confidence:** 4
**Clarity, Quality, Novelty And Reproducibility:** Comments related to clarity are addre…
**Correctness:** 3
**Technical Novelty And Significance:** 2
**Empirical Novelty And Significance:** 3
**Recommendation:** 3

**Strength And Weaknesses:**

Authors provide a good prior art section describing a range of papers that attempted the problem of developing new encoding formats to reduce the hardware cost of training high accuracy DNNs.

The paper tries to estimate hardware cost by estimating the approximate area of a design (adding the area for components misses glue logic) that trains DNNs for a given mantissa/exponent bit-width and block size. As a first order metric this is helpful to understand the savings achieved by reducing the mantissa bit-width or changing the block size. But a real hardware design would require the author to either choose a fixed HBFP format (e.g., HBFP5 only) or a few formats (e.g., HBFP4 and HBFP6) or to choose a reconfigurable hardware which will keep switching the size of HBFP mantissa bit-width used. Note that, the added cost of using reconfigurable hardware can be a limiting factor in its usage.

If the authors commit to having HW available to support a few formats, they can then simply compare the energy efficiency of using mixed HBFP format as a figure of merit.

Using the chosen FOM, approximate hardware area, Figure 1 illustrates the improvements achieved over FP32. To improve the readability of the figure, please add horizontal lines showing when the design achieves 95% of the hardware benefit. Also consider adding lines that link the points with same/similar accuracy for a given benchmark, e.g., 2% accuracy loss point for the different HBFP bit widths.
Authors highlight that HBFP6 achieves good accuracy for a range of benchmarks, this is a new empirical result not explored in prior art.

Section 4.1 explores the results of minimizing HBFP bit widths. This is a minor issue, but the authors claim that HBFP6 achieves accuracy within 2% of FP32 for all block sizes but ResNet50 trained for CIFAR-100 with block size 576 fails this condition. Some of the following discussion in this section is confusing, e.g., “for the smallest model ResNet20, decreasing the block size of HBFP5 to 25 leads to 1.5% accuracy degradation but 1.1× hardware gain in return”, it’s not clear what the baseline here is. Decreasing the block size generally helps improve accuracy and reduce the hardware gain. Also, it would be helpful for improving the readability if the authors added the area improvement ratios discussed in this section as a column in Table 1. It’s not easy to confirm these area improvement numbers from Figure 1.

Also, Table 1 highlights how using FP32 format for first and last layers helps improve the accuracy of training despite using HBFP format on other layers, but this has been done in previous papers. And the requirement of FP32 format even for a single layer means that Si area associated will be required, hence energy saving would be a better metric for comparing the efficacy of this technique.

Accuracy boosters is a useful technique which minimizes the HBFP mantissa bit width for epochs that are least impacted by it. The accuracy improvement achieved using this technique is remarkable. It enables HBFP4 to train ResNet50/74 to higher than FP32 accuracies which wasn’t feasible by default as show in Table 1. It is unclear however, that why the authors didn’t fully explore the benefit of using Accuracy boosters? How would it fare for HBFP3/2/1? They could potentially come with their tradeoff of how many epochs need a bit width boost.

Minor issue, the Conclusion section states, “Keeping the first and last layers in FP32 enables HBFP5 training with block sizes of 49 and higher for all models.”, it’s unclear what the authors mean here, cause for larger block sizes accuracy numbers drop regardless of the technique used.


**Summary Of The Paper:**

Block floating point format is a HW implementation technique for deep neural networks (DNNs), wherein floating point (FP) matrix multiply (Matmul) operations are simplified to fixed point multiply operations by sharing the exponent bits for a block of input values. Further, the mantissa bits for the fixed-point format are truncated to the minimum required for achieving the same or better accuracy for a given ML task than FP baseline.

This paper presents a design space exploration over mantissa bit-width and block size for a range of DNNs targeting different applications. Based on this exploration the authors identify that 6-bit mantissa format (HBFP6) achieve little or no loss of accuracy for significant HW savings.

Further, the authors propose the approach Accuracy Boosters, which uses the insight that the epochs in a model training have varying importance for final accuracy, e.g., last few epochs are critical and suffer the largest accuracy impact due to lower mantissa bit-width. Therefore, by selectively applying higher bit-width mantissa config (e.g., 6) to the last few epochs authors demonstrate a significant accuracy improvement even if majority of the cycles use lower mantissa bit-width (e.g., 4)

Authors show the accuracy results for a few DNNs targeting different applications and claim ~17x improvement in their figure of merit (FOM) for hardware cost, active Silicon area used.


**Summary Of The Review:**

The authors provide two new insights, 1) HBFP6 can achieve high accuracy for a range of networks and 2) Accuracy boosters is a reliable way to get higher accuracy while spending lesser energy. However, the figure of merit used by the authors, Si area, is not very useful as even a single usage of a unique data format requires associated hardware which if custom designed will come with its own area. Energy efficiency would have been a more useful metric. Also, the analysis of Accuracy boosters is limited and could use more experiments.

---

> ### Author Response · Authors · 2022-11-15
> **About the hardware implementation, adding power consumption numbers, addressing the other concerns and minor changes (Part 1/2)**
>
> Thank you very much for your valuable, detailed feedback and suggestions to improve our work. Below, we address each of your concerns and are happy to discuss further.
>
>
> > A real hardware design would require the author to either choose a fixed HBFP format (e.g., HBFP5 only) or a few formats (e.g., HBFP4 and HBFP6) or choose a reconfigurable hardware which will keep switching the size of HBFP mantissa bit-width used. Note that the added cost of using reconfigurable hardware can be a limiting factor in its usage.
>
> The preferred mixed precision scheme using mainly HBFP4 with a small number of HBFP6 epochs can be efficiently implemented with a minimal change in the hardware datapath, thereby not requiring fully reconfigurable ALUs. 6-bit multiplications can be emulated on 4-bit HBFP processing elements in four steps (cycles): take the four lower bits ($L$) and two higher bits ($H$) constituting the 6-bit mantissa. Then, compute the following intermediate products: $L* L$, $L * H$, $H * L$, and $H * H$, and accumulate them in the SIMD unit. Thanks to two's complement arithmetic, no modification to multipliers is needed. Only the accumulations in SIMD units need a constant shift of 4/8 bits, which can be provided using a single additional multiplexer in the exponent circuitry.
>
> > If the authors commit to having HW available to support a few formats, they can then simply compare the energy efficiency of using mixed HBFP format as a figure of merit.
>
> We have included a new table with the power savings of various HBFP configurations. The numbers show the ratio between the power consumption of FP32 hardware and various HBFP configurations for various models.
>
> > Figure 1 illustrates the improvements achieved over FP32. To improve the readability of the figure, please add horizontal lines showing when the design achieves 95% of the hardware benefit. Also, consider adding lines that link the points with same/similar accuracy for a given benchmark, e.g., 2% accuracy loss point for the different HBFP bit widths.
>
> We have added horizontal and vertical lines to Figure 1, showing when the design achieves 95% of the hardware benefit. We didn't add lines that link the points with the same/similar accuracy for a given benchmark because the figure became hard to follow. Instead, we included the area gains for each configuration in Table 1 as a column, as you suggested.
>
> > the authors claim that HBFP6 achieves accuracy within 2% of FP32 for all block sizes but ResNet50 trained for CIFAR-100 with block size 576 fails this condition.
>
> We explained the problem with the block size of 576 in the next sentence. However, the previous sentence needed to be clearer. So, we updated the sentence as the following:
> "HBFP6  is the smallest HBFP configuration that gives accuracies within  2%  of FP32 accuracy **for block sizes up to  256**. We also observe that unnecessarily large block sizes like  576  (providing only 0.8%  silicon area gain compared to a block size of  256) hurt accuracy for some models."
>
> > "for the smallest model ResNet20, decreasing the block size of HBFP5 to 25 leads to 1.5% accuracy degradation but 1.1× hardware gain in return", it's not clear what the baseline here is.
>
> This is compared to HBFP6. We updated the paper accordingly.
>
> > Also, Table 1 highlights how using FP32 format for first and last layers helps improve the accuracy of training despite using HBFP format on other layers, but this has been done in previous papers.
>
> Keeping the first and last layers in FP32 is not a novel idea. However, prior work does not investigate this method for block floating point formats. We don't claim that using full precision first and last layers is our contribution. It is a method that we included in our study to fully explore the potential of HBFP.
>
> > the requirement of FP32 format even for a single layer means that Si area associated will be required, hence energy saving would be a better metric for comparing the efficacy of this technique.
>
>  - Transformer models do not require the FP32 format so that they can benefit from a standalone accelerator designed for Accuracy Boosters. In the case of CNNs, we envision as one possibility that integrating HBFP4 accelerators -with support to HBFP6 through emulation- to a GPU architecture such as a part of the Tensor Cores can be beneficial for the energy efficiency of training. In this case, a small amount of FP32 computation can be carried out by the existing Tensor Core hardware.
>
>  - As per your suggestion, we included a new table with the power savings of various HBFP configurations (Table 4). The numbers show the ratio between the power consumption of FP32 hardware and various HBFP configurations for various models.

---

> > ### Author Response · Authors · 2022-11-15
> > **About the hardware implementation, adding power consumption numbers, addressing the other concerns and minor changes (Part 2/2)**
> >
> >
> > > It is unclear however, that why the authors didn't fully explore the benefit of using Accuracy boosters? How would it fare for HBFP3/2/1? They could potentially come with their tradeoff of how many epochs need a bit width boost.
> >
> > We already had experimented with HBFP3 and due to the extreme distortions that HBFP3 causes in the tensor distributions, it does not achieve enough accuracy. We have also tried investigating the tradeoff between the accuracy and the number of epochs using HBFP3. We trained ResNet20 on CIFAR10 with HBFP3 using the Accuracy Boosters of HBFP6. The accuracy numbers are given in the table:
> >
> > | Epochs using HBFP6 | Accuracy | Power saving over FP32 |
> > |--|--|--|
> > | Last 40 | 86.94 | 26.7x|
> > | Last 80 | 88.91| 21.9x|
> >
> >
> > Our preliminary results show that Accuracy Boosters is not working well with HBFP3. When we apply it for half of the training, the accuracy gets closer to FP32 accuracy (91.72). However, the power saving numbers show that using HBFP4 with the booster on the last epoch already provides more hardware benefits (28.2x).
> >
> > > Minor issue, the Conclusion section states, "Keeping the first and last layers in FP32 enables HBFP5 training with block sizes of 49 and higher for all models.", it's unclear what the authors mean here, cause for larger block sizes accuracy numbers drop regardless of the technique used.
> >
> > We show and state that HBFP6 is not sensitive to the block size, with the exception of unnecessarily large block sizes like 576, where the tensors obtained during training might be smaller than the block. However, HBFP5 starts showing more sensitivity to the block size. When we keep the first/last layers in FP32, this sensitivity decreases.

---

### Official Review · Reviewer_L7nb · 2022-10-29

**Confidence:** 4
**Correctness:** 2
**Technical Novelty And Significance:** 1
**Empirical Novelty And Significance:** 2
**Recommendation:** 3

**Clarity, Quality, Novelty And Reproducibility:**

It seems that many lessons suggested in this paper are incremental or well-known; e.g., full precision for the first/last layers, training with larger block size incurs more accuracy degradation, etc.). Therefore, the novelty of this paper is quite limited.


**Strength And Weaknesses:**

(Strengths)
- an interesting observation that the accuracy degradation by aggressively reduced mantissa bits can be recovered by the training with the increased number of mantissa bits.

(Weaknesses)
- This paper is mainly about the presentation of empirical findings, and many of them are already well-known (e.g., full precision for the first/last layers.)

- There are no deeper insights into the interplay between the block size and the mantissa bits.



**Summary Of The Paper:**

This paper investigated a proper bit-precision for a block floating-point format for deep neural network training and revealed that training with a small number of mantissa bits can be compensated if the training iterations with a large mantissa bit are followed. The authors also investigate the interplay between the block size and the mantissa bits for the block floating-point performance. The authors provided the experimental results to empirically support their claims.

**Summary Of The Review:**

In this paper, the authors investigated the characteristics of block floating point format. Although some of the empirical findings in this paper are interesting, many lessons derived from these observations seem to be incremental, limiting the novelty of this paper.

---

> ### Author Response · Authors · 2022-11-15
> **About novelty of our work and our insights**
>
> Thank you very much for your valuable comments. Below, we address each of your concerns and are happy to discuss them further.
>
> > This paper is mainly about the presentation of empirical findings, and  many of them are already well-known (e.g., full precision for the first/last layers.)
>
> We updated the introduction's second and last paragraph highlighting the novelty of our work and carried the paragraph about prior work in the introduction (which causes confusion/misunderstandings about our contributions) to the related work section.
>
> To the best of our knowledge, there is no prior work thoroughly investigating the parameter space of block floating point. We do not claim that using full precision first and last layers is our contribution. It is a method that we included in our study to explore the potential of HBFP fully.
>
> Accuracy Boosters is the first method that introduces high precision in the training process only for the last epoch and uses low precision for 99.7% of the training. Prior work using epoch- and/or layer-wise mixed-precision fails to make leaner arithmetic the common case for the training process. Our work shows, for the first time, that training DNNs with 4-bit arithmetic is possible with a last booster/fine-tuning touch.
>
> > There are no deeper insights into the interplay between the block size and the mantissa bits.
>
> We updated the first paragraph of Section 3.2, giving more mathematical explanation about our insight as the following:
>
> Block floating point shares a single exponent across a block of values using the exponent of the largest tensor value.
> Since block floating point format does not apply normalization ($2^{exponent}\times0.mantissa$ instead of $2^{exponent}\times1.mantissa$), the precision within a block is highly dependent on the largest element in that block, which decides on the exponent value.
> The interval between two consecutive representable numbers is calculated as follows:
>
>  $interval=\frac{2^{largest\ exponent}}{2^{number\ of\ mantissa\ bits}}$
>
> As the number of elements sharing the same exponent (block size) increases, the likelihood of disparity in the magnitude of elements also increases, leading to a precision loss for the small elements in the block.
> As the number of mantissa bits decreases, the model's sensitivity to the block size increases with the corresponding increase in the interval leading to a higher quantization error.
> More mantissa bits make the distribution more resilient to the quantization error and a larger block size, as each tensor element can be represented more accurately.

---

### Decision · Program_Chairs · 2023-01-20

**Decision:**

Reject

**Justification For Why Not Higher Score:**

Two reviewers voted strongly for rejection.

**Justification For Why Not Lower Score:**

N/A

**Metareview: Summary, Strengths And Weaknesses:**

Summary:

This paper investigated a proper bit-precision for a block floating-point format for deep neural network training and revealed that training with a small number of mantissa bits can be compensated if the training iterations with a large mantissa bit are followed. The authors also investigate the interplay between the block size and the mantissa bits for the block floating-point performance. The authors provided the experimental results to empirically support their claims.

Strengths:

- an interesting observation that the accuracy degradation by aggressively reduced mantissa bits can be recovered by the training with the increased number of mantissa bits.
- a good prior art section
- The authors provide two new insights, 1) HBFP6 can achieve high accuracy for a range of networks and 2) Accuracy boosters is a reliable way to get higher accuracy while spending lesser energy.
- Good background section that clearly explains the use of HBFP
- Intuitive explanation of mixed precision training in the context of HBFP with proper experimental backing
- Paper was easy to understand
- HBFP seems like a interesting data format with potential

Weaknesses:

- This paper is mainly about the presentation of empirical findings, and many of them are already well-known (e.g., full precision for the first/last layers.)
- There are no deeper insights into the interplay between the block size and the mantissa bits.
- the figure of merit used by the authors, Si area, is not very useful as even a single usage of a unique data format requires associated hardware which if custom designed will come with its own area. Energy efficiency would have been a more useful metric.
- the analysis of Accuracy boosters is limited and could use more experiments.
- It is unclear why training at the very end using HBFP6 allows you to recover the accuracy loss, more explanation would have been better
- The paper is entirely empirical. All conclusions are drawn largely from experiments on CIFAR-10/100 using ResNets and DenseNet40. One 1 table had data on transformers. This is a very weak set of benchmarks and models for an empirical paper.
- The results are not too exciting. For CNNs, the first and last layers must be kept in FP32 - so HBFP is only used for the "easy to quantize" layers. For the Accuracy Booster technique, it doesn't seem to work well on transformer.
- The paper combines two existing techniques: HBFP and FracTrain. The techniques are mostly orthogonal and not combined in a novel or insightful way.

Recommendation:

A majority of reviewers vote for rejection. I, therefore, recommend to reject the paper. I encourage the authors to use the feedback provided to improve the paper and resubmit to another venue.